# Health Studies in the Context of Artisanal and Small-Scale Mining: A Scoping Review

**DOI:** 10.3390/ijerph18041555

**Published:** 2021-02-06

**Authors:** Hermínio Cossa, Rahel Scheidegger, Andrea Leuenberger, Priska Ammann, Khátia Munguambe, Jürg Utzinger, Eusébio Macete, Mirko S. Winkler

**Affiliations:** 1Swiss Tropical and Public Health Institute, P.O. Box, CH-4002 Basel, Switzerland; andrea.leuenberger@swisstph.ch (A.L.); pammann9392@gmail.com (P.A.); juerg.utzinger@swisstph.ch (J.U.); mirko.winkler@swisstph.ch (M.S.W.); 2University of Basel, P.O. Box, CH-4003 Basel, Switzerland; 3Manhiça Health Research Centre, C.P. 1929 Maputo, Mozambique; khatia.munguambe@manhica.net (K.M.); eusebio.macete@manhica.net (E.M.); 4Swiss Federal Institute of Technology, P.O. Box, CH-8092 Zurich, Switzerland; rahel.scheidegger@gmx.ch; 5Faculty of Medicine, University Eduardo Mondlane, C.P. 257 Maputo, Mozambique; 6National Directorate of Public Health, Ministry of Health, C.P. 264 Maputo, Mozambique

**Keywords:** artisanal and small-scale mining, health effects, health hazards, low- and middle-income countries, mercury, injuries and fatalities

## Abstract

Artisanal and small-scale mining (ASM) is an important livelihood activity in many low- and middle-income countries. It is widely acknowledged that there are a myriad of health risk and opportunities associated with ASM. However, little is known with regard to which aspects of health have been studied in ASM settings. We conducted a scoping review of peer-reviewed publications, using readily available electronic databases (i.e., PubMed, Scopus, and Web of Science) from inception to 14 July 2020. Relevant information was synthesized with an emphasis on human and environmental exposures and health effects in a context of ASM. Our search yielded 2764 records. After systematic screening, 176 health studies from 38 countries were retained for final analysis. Most of the studies (*n* = 155) focused on health in ASM extracting gold. While many of the studies included the collection of environmental and human samples (*n* = 154), only few (*n* = 30) investigated infectious diseases. Little attention was given to vulnerable groups, such as women of reproductive age and children. Our scoping review provides a detailed characterisation of health studies in ASM contexts. Future research in ASM settings should address health more comprehensively, including the potential spread of infectious diseases, and effects on mental health and well-being.

## 1. Introduction

Artisanal and small-scale mining (ASM) is commonly defined as any mining activity of ore deposits that are considered not cost-effective for large-scale industrial extraction [1]. Although cross-country distinctions between artisanal mining and small-scale mining exist, both share a number of common characteristics [2]. For example, both artisanal mining and small-scale mining are activities predominantly pursued by the informal sector, usually involving a few to hundreds of individuals using basic equipment [3]. Consideration of health and safety measures in ASM working processes are low [3,4,5,6]. It is currently estimated that ASM activities are practiced by more than 40 million people in over 120 countries [4,7,8]. Most of the activities are concentrated in low- and middle-income countries (LMICs) in Africa, Asia, and the Americas [4,5]. Driven by a number of factors (e.g., demand for minerals, gold price, and poverty), the ASM sector shows a significant diversity in various dimensions: (i) demography (up to 50% of miners are women and 10% are children); (ii) origin of miners (mix of local communities and migrant mine workers); (iii) seasonality (e.g., rush mining due to discovery of new mineral deposits, resulting in rapid in-migration); (iv) legal status (e.g., 70–80% of ASM are informal in many countries); and (v) targeted commodities (e.g., gold, cobalt, coltan, and diamond) [7,8,9]. Despite the informal nature of activities, ASM represents a considerable share in global minerals and gemstones extraction. For instance, approximately half of the ASM miners are working in gold mines with a share of 20% of the global gold supply [2,8]. In addition to gold, ASM has a considerable global share in the supply of sapphire (80%), tantalum (26%), tin (25%), and diamonds (20%) [8]. Taken together, ASM is an important economic activity that provides livelihood opportunities for an estimated 80–150 million people [10,11].

Worldwide, ASM activities are on the rise. The activities are associated with both risks and opportunities for the health of miners, their families, surrounding communities, and populations living up- and downstream from ASM sites [2,5,12]. For instance, ASM can induce changes in people’s socio-economic status and potentially reduce poverty, and improve their ability to access health care and education [13,14]. Income from ASM employment may reduce food insecurity [15] and improve housing conditions, which may result in reduced transmission of vector-borne diseases, such as malaria [16] and improved respiratory health [17]. Furthermore, it has been suggested that health and environmental benefits may be achieved, should governments and other key players formalise the ASM sector, provide financial assistance, promote improved technologies, educate on environmental protection, and encourage artisanal miners to form associations [18,19].

However, positive effects on health associated with ASM activities are often overshadowed by negative impacts. It is interesting to note that concerns about risks associated with occupational exposures in mining settings have already been raised in the 17th century by Ramazzini [20]. More than two centuries later, the exposure of surrounding communities to diseases related to mining was reaffirmed by Rose [21] and Cole [22]. Today, ASM is still expanding and generally presents as a highly labour-intensive activity with low health and safety standards in place [23], exposing the ASM workforce—often including women of reproductive age and children [6,9,12]—to a range of chemical, physical, biological, biomechanical, and psychosocial hazards [5,12,24], frequently resulting in “rapid development of disease and premature death” [21]. In addition to the occupational environment, ASM-related health hazards also affect surrounding communities, including particularly vulnerable groups (e.g., children, women of reproductive age, and elderly) [2,8,25]. Indeed, health effects related to chemical exposure, such as mercury (Hg), cobalt (Co), arsenic (As), and lead (Pb), including neurological, renal, and autoimmune disorders, have been reported in occupationally and non-occupationally exposed individuals in ASM settings [24,26,27,28,29,30,31,32]. Moreover, mining-induced in-migration and changing life styles pose considerable health risks affecting miners and exposed communities [2,8,12,33,34].

Taken together, it is widely acknowledged that ASM activities affect human health in myriad direct and indirect ways. However, little is known about how exactly people’s health and well-being are affected by ASM contexts in different parts of the world. To our knowledge, most health-related research on ASM settings have focused on gold extraction (i.e., ASGM) and associated exposures to Hg [23,35,36,37,38,39]. Indeed, a considerable body of evidence is available on environmental and human exposure to Hg in ASGM contexts [24,29,35]. At the same time, many other health hazards and risks are likely to prevail in the context of ASM that received far less attention [40]. Similarly, health risks that are ubiquitous across any type of ASM might have been studied more frequently in ASGM context, as compared to effects that are specific to other minerals being extracted in ASM [23]. For example, few studies pertaining to health effects in ASM contexts have tempted to include exploitation other than gold, such as artisanal fluvial extraction of river sand and informal gasoline trade [40]. An overview of health studies in the context of ASM will not only enhance the understanding of which health aspects have been studied thus far, but will also allow to identify potential research gaps that can guide future research efforts in ASM.

This paper aims to provide a global overview of human health-related studies that have been carried out in ASM contexts, placing particular emphasis on those directly engaged in ASM activities, as well as those living within mining areas and surrounding populations. The research was guided by the following questions: (i) in which countries have ASM activities been conducted and what type of ASM contexts have been studied? (ii) What health effects and ASM-related exposures to health hazards (environmental or biomarkers) were studied? (iii) Which population groups were included in these studies? (iv) What type of environmental and human samples were collected? (v) What type of exposures were measured? (vi) Which signs, symptoms, diseases, injuries, and fatalities were investigated?

## 2. Materials and Methods

A scoping review was conducted targeting the peer-reviewed literature [41]. The search was oriented towards, but not limited to, epidemiological studies that investigated health issues such as communicable and non-communicable diseases, signs and symptoms, injuries, and fatalities in ASM contexts. Established health indices such as human biomonitoring (HBM), health risk assessment (HRA), and risk analysis (RA) that were applied in ASM settings, were also considered. Studies that were carried out in industrial mining contexts were deliberately not included in the current scoping review, as our focus was on ASM.

### 2.1. Search Terms and Strategy

Relevant studies were identified through a systematic search guided by the “Preferred Reporting Items for Systematic reviews and Meta-Analyses extension for Scoping Reviews (PRISMA-ScR): Checklist and Explanation” [42]. Two steps were taken to develop the search terminology. First, the search terms were independently developed as an iterative process by two reviewers (H.C. and R.S.). Second, the two draft search terms were compared and consolidated into a final search strategy. Any disagreement was discussed with another two researchers (A.L. and M.S.W.). The final search strategy (in English only) consisted of two search term blocks: (i) ASMs related and (ii) health effects related terms (see Appendix A, Table A1).

### 2.2. Peer-Reviewed Literature Searches and Screening

We conducted the electronic literature search applying a systematic search strategy in three readily available electronic databases, namely (i) PubMed; (ii) Scopus; and (iii) Web of Science (WoS). No spatial, temporal, or language restriction was applied for the search term administered on 14 July 2020. The records retrieved from the three databases were pooled together and imported into EndNote version X9.2 (Thomson Reuters Corp.; New York City, NY, USA; https://endnote.com). Of note, the search strategy was amended to the specific features of the databases (see Appendix A, Table A1).

In a first step, records were de-duplicated based on authorship, title, and publication year of the articles identified in the different databases. For this purpose, automatic detection and hand curation was done using both EndNote and Microsoft Excel (Microsoft Office Standard 2016, version 16.0.4266.1001; Microsoft Corporation, Redmond, WA, USA). In a second step, all records other than original research papers and reviews were excluded (i.e., conference proceedings, books, book chapters, editorials, opinion pieces, patents, and correspondences) using the reference type field in the EndNote software. In a third step, titles and abstracts of all records were independently screened by four reviewers, including three authors (H.C., P.A., and R.S.) and one collaborator (A.G.) using EndNote and Microsoft Excel for data management. Discrepancies were discussed in pairs (H.C.-R.S., H.C.-A.G., and H.C.-P.A.) until consensus was reached, if need be with input of an additional author (M.S.W.).

The remaining full texts were screened uniquely by H.C. with support, whenever needed, from M.S.W., applying the following set of inclusion criteria for the final selection of the relevant articles: (i) study was carried out in an ASM environment or an environment affected by ASM; (ii) study investigated human health outcomes and/or human health related indicators (environmental samples or human biomarkers); (iii) the articles is open access or accessible with the rights of the University of Basel (Basel, Switzerland). A post hoc approach for inclusion criteria was applied in order to focus on the comparison of quantitative epidemiological studies solely conducted in the context of ASM. Hence, qualitative studies, systematic reviews, and studies focusing on both ASM and other (illegal/informal) activities, such as farming and electronic waste recycling, were excluded for the current scoping review.

### 2.3. Data Extraction and Analysis

Two authors (H.C. and M.S.W.) developed the data charting form in a Microsoft Excel spreadsheet for data extraction with the following variables: articles’ background data (author, year of publication, country, study type, study design, and natural resources extracted); characteristics of study population (age, age group, sex, and population groups); type of health-related samples taken (environmental samples and related pollutants, and human samples and related biomarkers); investigated health issues (signs and symptoms related to chemical exposures; adverse health conditions, and fatalities); and health risk indices (HRI), i.e., health indicators calculated to quantify human health risks associated with the environmental hazards [43,44]. Online electronic supplementary materials of included articles were accessed and screened for additional data extraction whenever necessary. Data extraction based on the full-text analysis was mostly driven by one researcher (H.C.). Microsoft Excel (Microsoft Office Standard 2016, version 16.0.4266.1001, Microsoft Corporation; Redmond, WA, USA) was used for data entry and cleaning, and subsequently imported into STATA (Stata Corporation, version 14.2, LLC; College Station, TX, USA) for data analysis. Drawing on the concept put forth by Levac and colleagues [41], a descriptive thematic approach was used to characterise the included articles. Identified categories were summarised as frequencies and continuous variables as median and interquartile range (IQR). In cases where the characterisation of the articles refers to a sub-group of the total number of articles identified, the nominator (number of articles of interest [*x*]), denominator (total number of relevant articles [*y*]), and percentage (%; [*x/y*]) are specified in the results section. The number of studies per country were compiled using Microsoft Excel and exported into QGIS (version 3.14.0-Pi, Free Software Foundation, Inc.; Boston, MA, USA; https://qgis.org/en/site/) for illustration. In case multiple countries were reported in a single article, countries were counted separately. Similarly, the type of extracted commodities and number of reporting studies were compiled using Microsoft Excel. When more than one commodity was reported, a separate counting was done for each. PowerPoint (Microsoft Office Standard 2016, version 16.0.4266.1001, Microsoft Corporation; Redmond, WA, USA), was used for edition and final illustration of all figures.

## 3. Results

### 3.1. Overview of Studies

We identified a total of 2764 records based on PubMed, Scopus, and WoS. After de-duplication (1176 removed records), 1588 unique studies remained from which 1412 were excluded based on reference type (*n* = 289), title and abstract screening (*n* = 987), and full text screening (*n* = 136). Hence, 176 articles were deemed eligible for data extraction and analysis, as shown in the PRISMA flow chart (Figure 1). All articles included were in English with the exception of two that were in Spanish.

### 3.2. Study Characterization Regarding Country and Year of Publication

Panel A in Figure 2 shows the geographical distribution of the 176 included articles that present primary (*n* = 167) and secondary (*n* = 26) data from 38 countries. Most of the studies were conducted in Africa (*n* = 95; 16 countries), followed by Asia (*n* = 53; 10 countries) and Latin America and the Caribbean (*n* = 50; 11 countries). Only one study involved individuals from a European country [45]. The number of published articles per country ranged from 1 to 32. Ghana is the country with the largest number of published articles (*n* = 32).

While most of the studies focused on a single country (93.8% [165/176]), studies including more than one country (i.e., multi-centric studies) accounted for 6.2% [11/176]. Multi-centric studies involved two to five countries from Africa (*n* = 2), Africa and Asia (*n* = 8), and South America and Europe (*n* = 1).

The oldest paper identified was published in 1998 (Figure 2, Panel B). The median (IQR) of articles per year was 5 (range: 2–13). Two thirds of the articles (67% [118/176]) were published between 2015 and 2020, indicating an increasing trend in peer-reviewed ASM-related published articles in the last 5 years.

### 3.3. Study Characteristics Regarding Key Topics Covered

Of the 176 included health studies, most were related to HBM (*n* = 32), HRA (*n* = 30), and human health assessment (*n* = 26). Few studies (*n* = 4) presented medical case reports (Figure 3). The remaining studies were related to integrated approaches, comprising HBM and health assessment (HA) (*n* = 68), HA and HRA (*n* = 9), HBM, HRA and HA (*n* = 6) and HRA and HBM (*n* = 1). Of note, three out of four medical case reports were related to both HBM and HA [47,48,49].

A high diversity of types of health studies was observed (Figure 4). Based on the extracted data, we examined how included studies (*n* = 176) assessed health-related exposure and health effects from ASM activities. The following study type clusters were identified: (i) HA (i.e., medical investigation, *n* = 113); (ii) HBM (*n* = 110); (iii) HRA studies (*n* = 46); and (iv) medical case reports (*n* = 4). A large number of articles (*n* = 71) applied integrated environmental monitoring (EM) approaches, applying different combinations of HBM, HA, and HRA.

Exposure assessments were performed through examination of human samples, environmental samples (including biotic and food samples), and self-reported exposure through questionnaires. In all studies, samples were submitted to laboratory examination for chemical and biological hazards exposure assessment. Physical hazards exposure was mainly assessed through questionnaires and field observations. Of note, most of HRA studies (*n* = 45) investigated chemical hazard exposure, including heavy metals, metalloids, and trace elements.

In regards to the strategies applied for investigating health outcomes from ASM exposure, a broad range of approaches were employed (Figure 4). For instance, HBM studies were more focused on the level of internal body exposure to chemical hazards (e.g., Hg). Signs, symptoms, diseases, injuries, and fatalities were reported in the remaining study type clusters, except for HRA studies, which reported health risk indices. The main tools used to investigate health outcomes and other health-related indicators were: (i) laboratory analysis of samples (for pollutants, infections and physiologic parameters); (ii) questionnaires for self-reported health outcomes and health risk indices estimations using HRA tools; and (iii) medical examination.

#### 3.3.1. Type of ASM as Function of Extracted Commodities

Figure 5 shows the type and number of commodities reported to be extracted in the ASM contexts investigated in the included articles. A total of 23 extracted commodities were reported. The large majority of identified studies were carried out in ASGM context (*n* = 155). Other less reported commodities, such as coltan, aluminium (Al), coal, diamond, lead (Pb), Hg, and tin (Sn) are also shown in Figure 5. Of note, 10 studies were conducted in ASM settings where more than one commodity was extracted, including gold (Au), diamond, and emerald, (*n* = 2) [50,51]; iron (Fe), Sn, and Al (*n* = 2) [44,52]; Pb and zinc (Zn) (*n* = 2) [53,54]; Au, limestone, and Al (*n* = 1) [55]; Au and copper (Cu) (*n* = 1) [56]; Au, Sn, cassiterite, coltan, and gemstones (*n* = 1) [57]; and Au, manganese (Mn), and bauxite (*n* = 1) [58].

#### 3.3.2. Characteristics of Study Populations

All studies, except one [59], mentioned the gender of the study population (Table 1). Most (87.5% [154/176]) included males and females. Of those, half (50.6%; [75/152]) included individuals of all ages (i.e., from 6 months and above) followed by those including adults only (i.e., 18 years and above) (32.9% [50/152]). More than half of these studies (54.5% [6/11]) investigating health in female subjects only, were focused on specific subgroups, i.e., pregnant women (*n* = 5) [32,60,61,62,63] and women of reproductive age (*n* = 1) [64].

As shown in Table 1, almost half of the studies (48.3% [85/176]) included both miners and residents. Two-thirds of these studies (67.0% 57/85]) addressed population groups of all ages and a quarter (25.0%; [21/85]) focused on adults (aged ≥18 years) only. Few studies (5.7% [10/176]) investigated children’s health. Seven articles (4.0% [7/176]) did not mention the age of study subjects. The maximum reported age was 98 years [65]. Among those investigating health in children, one included children working in ASM [59].

### 3.4. Type of Investigated Samples and Health Effects

Overall, 154 studies were identified that analysed at least one type of environmental or human samples. Human biological samples were collected in 114 studies. Environmental samples of human health relevance were collected in 71 studies. Thirty-one studies comprised both environmental and human sampling. While Hg alone was investigated in 74.4% [131/176] of the studies, physical (e.g., dust, exposition, and landslide) and biological hazards (e.g., human immunodeficiency virus (HIV), malaria, and soil-transmitted helminth infections) were investigated in 27 and 18 studies, respectively.

#### 3.4.1. Environmental Samples and Related Pollutants

In all included studies (*n* = 176), 10 different types of environmental samples were collected and analysed. Soil (*n* = 36), water (*n* = 35), fish (*n* = 22), sediment (*n* = 21), and non-fish food (*n* = 15) were the most frequently collected environmental samples (Figure 6, Panel A).

Hg and Pb were the most investigated ASM-related pollutants. For instance, in water samples Hg and Pb concentration levels were assessed in 24 and 14 studies, respectively. The same pattern was observed for other environmental samples (Figure 6, Panel B). The highest diversity of assessed environmental pollutants was observed in soil samples (28 pollutants). Air samples were analysed exclusively for Hg concentration using both portable Hg detector devices [49,66] and laboratory air analysis [67]. One study investigating air pollution measured oxygen (O_2_) and carbon monoxide (CO) concentrations, and other physical parameters, such as temperature and dust concentration [68]. Other environmental pollutants reported in a few studies investigating soil, water, fish, sediment, and food samples are summarised in Appendix B, Table A2.

As part of Appendix B, Table A3 provides details on environmental pollutants determined in dust and ore samples and Figure A1 and Figure A2 give details on other environmental samples, such as non-eatable plants and tailings, respectively.

#### 3.4.2. Human Samples and Related Biomarkers

Panel A in Figure 7 presents the different types of human samples and biomarkers (internal body exposure) investigated in the included studies. Overall, 11 types of human samples were collected and analysed for both exposure assessments and medical examinations of study participants. The three most investigated types of human samples are hair (*n* = 71), urine (*n* = 54), and blood (*n* = 51).

The diversity of pollutants assessed in hair (*n* = 46) was higher than in urine (*n* = 25) and blood (*n* = 13) (Figure 7, Panel B). Most biomarkers were reported in less than four articles each. Two studies contributed significantly to the diversity based on the assessment of hair-biomarkers [69,70]. Similarly, the diversity of pollutants revealed through urine biomarkers was mostly related to the study from Nkulu and colleagues [31], measuring the level of 25 different pollutants in urine. Hg levels measured in hair (*n* = 69), urine (*n* = 50), and blood (*n* = 36) were clearly the chemical exposure most assessed in the last two decades. The second most reported pollutant was the level of Pb (measured in blood, *n* = 8) and arsenic (measured in urine, *n* = 6). Moreover, breast milk samples were exclusively analysed for Hg concentration levels [71,72]. Other less reported chemicals are given in more detail for hair, urine, blood, and nail biomarkers in Appendix C, Table A4. The list and frequencies of physiological parameter measured in serum is provided in Appendix C, Table A5.

Besides chemical pollutants, blood, and urine samples were also analysed for other physiological parameters, such as blood lactate and cyanide [73,74], haematological parameters, and other blood chemistry [47,75], mitochondrial DNA, and other DNA mutations [48,76,77,78], immunological response to vaccines, and other immune-related proteins [50,79,80], renal and respiratory parameters [75,81], and infectious diseases, such as malaria [82] and HIV [47,83]. Saliva (cortisol salivary) [84], Buccal cells (DNA damage and mutations) [63,77], and stool samples (soil-transmitted helminths and *Vibrio cholerae*) [85,86] were also reported.

### 3.5. Adverse Health Conditions and Fatalities

Of the 176 studies included in our analysis, 144 (81.8%) reported at least one of the following three groups of health issues: (i) prevalence or incidence of diseases and other adverse health conditions (*n* = 97); (ii) signs and symptoms related to chemical exposures (*n* = 71), with Hg exposure being studied most often (*n* = 53); and (iii) musculoskeletal disorders, injuries, and fatalities (*n* = 37) (Figure 8). An integrated approach was applied in more than half of the studies (*n* = 74). For instance, a combination of two (*n* = 48) and three (*n* = 20) types of health issues were identified and six studies incorporated all four types of health issues [87,88,89,90,91] in their reports. Health effects from gold extraction activities alone were investigated in 84% [149/176] of the studies.

#### 3.5.1. Signs and Symptoms Related to Chemical Exposure

The use of Hg and its health effects were the major issues in the studies that met our inclusion criteria. We identified 71 studies addressing at least one sign and symptom related to exposure to Hg and other metals. Notably, most of these studies (61.9% [44/71]) had a strong focus on Hg alone, while an additional 12.7% [9/71] focused on concomitant exposure to Hg and other metals, such as Pb and As. Two-thirds of the studies [47/71] investigated potential signs and symptoms of exposure to environmental contaminants. Forty-one out of 44 studies investigating neurological symptoms assessed Hg exposure in the affected population. Likewise, 21 and 19 studies of those investigating visual disorders (*n* = 27) and neuro-motor function disorders (*n* = 21) assessed Hg exposure, respectively. All studies reporting neuropsychological symptoms (*n* = 20) also assessed Hg exposure. Other signs and symptoms included pain, hearing disorders, stress, cognitive impairment, and intelligence quotient (Figure 8). Only three studies investigating signs and symptoms of cyanide exposure were identified [73,74,92].

#### 3.5.2. Adverse Health Conditions

Our study sample revealed 34 subcategories of diseases and other adverse health conditions. As shown in Figure 8, the six most frequently studied subcategories were: (i) nutritional disorders (*n* = 38); (ii) respiratory disorders (*n* = 32); (iii) renal diseases (*n* = 31); (iv) gastrointestinal disorders (*n* = 29); (v) poisoning (*n* = 29); and (vi) cardiovascular diseases (*n* = 23). Thirty studies addressed at least one infectious disease, such as malaria (*n* = 15), tuberculosis (*n* = 11), HIV infections and STIs (*n* = 7), typhoid (*n* = 5), and pneumonia (*n* = 4). Details on other types of communicable diseases reported in less than four studies are provided in Table A6, Appendix D.

#### 3.5.3. Musculoskeletal Disorders, Injuries, and Fatalities

Eleven types of musculoskeletal disorders, injuries, and fatalities were reported in 37 studies. Musculoskeletal disorders (*n* = 17) were most frequently reported, followed by wounds (*n* = 11) and burns or abrasions (*n* = 11). Fatalities were reported in 16 studies, of which 12 were conducted in African countries.

### 3.6. Health Risk Indices

As shown in Figure 9, five HRIs were reported in 46 studies out of 176, namely (i) hazard quotient (HQ); (ii) hazard index (HI); (iii) cancer risk (CR); (iv) permissible exposure limit (PEL); and (v) risk rating (RR). The most frequent health index was the HQ (*n* = 36) of which (88.9% [32/36]) were from studies assessing chemical exposure from environmental samples only. Both total and targeted HQ (T-HQ) were used in some studies. HI is the sum of all computed HQs [93] and was computed in 20 studies. Both HQ and HI are individual health risk estimation for non-carcinogenic endpoint of environmental contaminant exposure [94].

The following most estimated HRI was the CR (*n* = 21), which is a term used to characterise lifetime probability of developing any type of cancer disease [52]. Two synonyms are used for this estimate, such as incremental lifetime cancer risk (ILCR) and incremental excess lifetime cancer risk (IELCR) [94,95]. This indicator was exclusively estimated by human health risk assessment studies. RR (*n* = 8) and PEL (*n* = 3) were the less estimated HRI.

Adding to the five previous estimates, the intake rate estimate (i.e., chronic daily intake, CDI) were also considered in about a third of the studies (*n* = 55). Three routes of pollutants intake were reported. Ingestion (*n* = 53) was reported most frequently, followed by inhalation (*n* = 18) and dermal absorption (*n* = 16). Furthermore, 12 studies considered the three intake routes in an integrated manner.

## 4. Discussion

A total of 176 articles from 38 countries in Africa, Asia, and the Americas were identified in our scoping review and subjected to in-depth analysis. Systematic classification of the studies revealed a heavy focus on health research in the context of ASGM. Consequently, Hg exposure and associated health-related effects in affected populations were by far the most investigated research topics. The transmission of infectious diseases, such as malaria, STIs (mainly HIV), and tuberculosis in ASM contexts has received comparatively little attention. Musculoskeletal disorders, wounds, and burns were the types of injuries mostly investigated. Only a few health studies specifically investigated vulnerable population groups, such as women of reproductive age and children.

### 4.1. Representativeness of Health Research in ASM

ASM is practiced in 124 countries, primarily in LMICs in Africa (*n* = 49 countries), Asia (*n* = 35 countries), and Latin America and the Caribbean (*n* = 29 countries). In Europe (*n* = 6 countries), Northern America (*n* = 3 countries), and Oceania (*n* = 2 countries), ASM is far less often pursued [4]. However, the 176 health-related research articles included in our scoping review represent only a third of the countries hosting ASM activities (37.9% in Latin America and the Caribbean, 33.7% in Africa, 28.6% in Asia, and 16.7% in Europe). None of the included studies were conducted in Northern America and Oceania. Hence, from two-thirds of the countries with ASM activities, no health-related research has been reported in the peer-reviewed literature. When considering that ASM contexts are hotspots for occupational risks and public health challenges [23,24], this finding clearly points out that health research in ASM is a neglected area of research in many countries hosting ASM activities.

There are several potential underlying reasons why health in the context of ASM lacks pointed research. First, despite the current trend in urban mining, such as gold jewellery manufactures [96], ASM operations are often performed in distant or difficult-to-access settings [2,8]. Hence, data collections at mining sites are challenging in terms of access, while financial and human resources for research are often constrained [35,36]. Second, ASM activities are generally informal and take place in contexts with high cultural sensitivities [8,97,98]. This poses challenges to accessing people involved in ASM, including vulnerable population groups such as women of reproductive age, children, and ethnic minorities. Third, research efforts that aimed to build on secondary data sources from ASM settings have faced challenges with data availability and quality. For instance, it was found that data availability and quality were limited for estimating disease burden and disability-adjusted life years (DALYs) due to Hg use in ASGM in Zimbabwe and elsewhere [35,36]. Despite the many potential challenges faced with the implementation of health studies in ASM settings, the high public health relevance—more than 40 million being directly exposed to ASM [4]; many are vulnerable or marginalised groups [8]; and many potential health issues [8,12]—calls for the definition of a comprehensive global research agenda that promotes health research in ASM contexts. This also aligns with ongoing initiatives such as the Global Mercury Partnership and the agenda of the World Bank, which has already recognised this topic as a “big global data gap” [15].

### 4.2. Inclusion of Vulnerable Groups

In health studies carried out in ASM contexts, women of reproductive age and children were rarely investigated as stand-alone groups. Among the few studies identified that included vulnerable groups, none focused specifically on the health status of women of reproductive age or children who were directly involved in ASM activities. Hence, health research in ASM contexts largely missed out on the most vulnerable population groups. This is in contrast to existing literature that has long established the vulnerability of women of reproductive age and children in ASM [2,5,12,99]. The limited health data pertaining to these vulnerable groups may be due to fear of ethical issues when conducting research involving vulnerable population groups [100], the informal nature of the sector [12], as well as cultural sensitivities and high gender-based inequities [97], which renders these groups less reachable due to their fear and distrust of participating in research studies [101]. When considering that an estimated 30–50% women and more than one million children are directly involved in ASM activities [15], better targeting, and inclusion of vulnerable groups in health studies carried out in ASM communities is a pressing need [102].

In order to address these challenges, it is essential that ethical boards, national and local health authorities, local organisations working in ASM settings and, most importantly, ASM communities are involved in every phase of the research process [103]. Indeed, early involvement of ethical boards presents an opportunity for overcoming potential uncertainties on how to invite vulnerable individuals to serve as participants in health research in ASM contexts with all the required special justification [100,104]. The integration and support from multi-sectoral and multidisciplinary stakeholders, including collaboration with governmental and non-governmental organisations, indigenous populations, local leaders, and potential participants is a common practice and proven useful [49,105,106,107]. The value added of such multi-stakeholder engagement enhances trust between participants and scientists when conducting research involving vulnerable groups [100,101,105]. Independent of the context and chosen approach, it is critical to give the community´s voices and decisions equal importance in addressing issues related to limited health data on vulnerable groups [49,106].

### 4.3. Gaps Identified in Exposure Assessments

The systematic characterisation of the articles identified in our scoping review revealed a high diversity of exposure assessment approaches, comprising the sampling and analysis of 51 different chemical hazards in both environmental and human samples. While the systematic use of multiple biomarkers when assessing the effect of (environmental) pollutants has been suggested previously [108], the observed strong focus on Hg is striking. This finding may, at least partially, be explained by the adoption of the Minamata Convention on Mercury in 2013, which triggered considerable interest in Hg pollution and exposure in ASGM [102,109]. Indeed, three-quarters of the articles identified were published after the adoption of the Minamata Convention in 2013. At the same time, the emphasis on Hg use in ASGM may be overshadowing other important health issues that prevail in ASM communities, such as silicosis and tuberculosis, water- and vector-borne disease and sexual health issues, such as STIs, including HIV [23]. This is confirmed by a global study showing that ASM scholars and practitioners are more concerned with Hg exposure in ASM communities than other health hazards [14]. Indeed, Hg exposure and its toxicological consequences pose considerable health and environmental concerns due to the expansion of ASGM activities into urban and sub-urban communities [96,110]. With only one quarter of the health studies in ASM contexts that addressed other health issues than Hg exposure in ASGM, there might be an important research gap. Future studies should therefore make an effort to comparatively assess the public health relevance of different chemicals, biological, biomechanical, physical, and psychosocial hazards that occur simultaneously in ASM settings [2,5,16], for ultimately developing a research agenda that addresses all major health risks and opportunities in ASM.

### 4.4. Widening the Focus of Health Conditions Studied

Less than one in five articles identified studied communicable diseases, such as malaria, tuberculosis, and HIV in ASM contexts. When considering that 68.8% of the burden of disease in LMICs, where ASM is mostly practiced, is caused by communicable, maternal, neonatal, and nutritional diseases [111,112], this finding is surprising. Moreover, it is likely that these conditions are more prevalent in ASM settings compared to the countries’ average due to the remote environments and potentially weak local health systems [111,113]. Indeed, studies have shown that ASM settings are characterised by high prevalence of air-, water-, blood-, and vector-borne diseases [5,114,115], as well as STIs and HIV [5]. The observed “high polarity” in health-related studies in ASM context may be linked to limited concern on other health hazards beyond Hg among ASM experts worldwide [14]. As a consequence, today, we face a systematic health data gap in the ASM sector at both national and regional levels, and, thus, scarce knowledge about the existing relationship between ASM activities, and the effect of other harmful exposures to the environmental and human health; a situation that may even further deteriorate in the current and future scenario of the coronavirus disease 2019 (COVID-19) pandemic.

### 4.5. Strengths and Limitations

This scoping review attempted to draw a general picture of research covered, thus far, with regard to health issues in ASM communities worldwide. Given the unrestricting nature of our piece (i.e., all types of ASM were included), we feel that our study makes a meaningful contribution to the literature, providing new insights into health research practice in ASM contexts. This scoping review, however, comes with several limitations. First, we did not include grey literature and peer-reviewed articles for which no titles and abstracts in English were available. Second, radiological studies, systematic reviews, and qualitative studies were, post hoc, excluded from data extraction and analysis due to time and human resources constraints. Third, due to the lack of standard terms for investigated adverse health conditions, such as symptoms of Hg intoxication among included studies [27,63,82,116], misclassification or entry duplication may have occurred during data extraction, which may inflect or deflect our findings. Fourth, we did not characterise the included articles in relation to the location where studies were conducted (e.g., rural vs. urban mining), which might have affected the interpretation of our results [45,49,96,110]. Fifth, we acknowledge that for conclusively judging the inclusion of vulnerable populations groups (e.g., women of reproductive age, children, and ethnic minorities) in health research in ASM contexts, a more holistic review is needed that does not only focus on published peer-reviewed literature.

## 5. Conclusions

Heath studies in ASM contexts have gained in popularity and diversity over the past two decades. At the same time, when considering the global magnitude of ASM practice, the number of health-related studies in ASM appears to be low and several specific contexts are not adequately represented in the existing body of literature. This also applies to ASM activities other than ASGM. Indeed, 88.1% of the articles identified are representing gold mining contexts, and, thus, other ASM commodities, such as coltan and cobalt, both showing increasing demand worldwide, are largely neglected. Furthermore, there is a dearth of studies that have focused on specific vulnerable population groups that are directly involved in ASM processes. In addition to a biased representation of ASM contexts and affected population groups, health studies in ASM communities show a heavy focus on Hg-related exposures and associated health effects, whilst communicable diseases and other health conditions not associated with Hg use have received relatively little attention. Future research efforts addressing the “global health data gap” in ASM contexts should diversify in terms of commodities and health issues studies, while also paying particular attention to the most vulnerable population groups.

## Figures and Tables

**Figure 1 ijerph-18-01555-f001:**
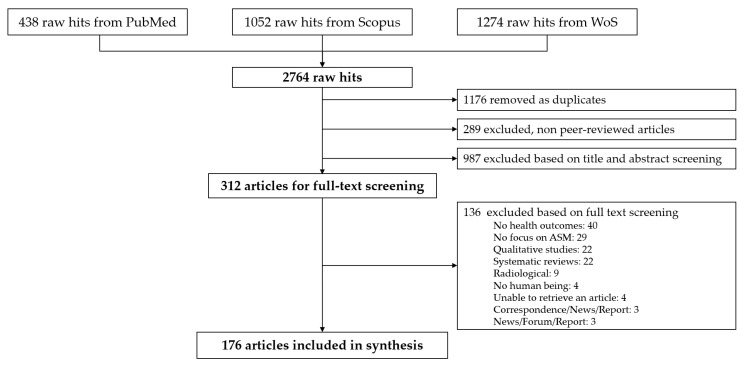
Preferred Reporting Items for Systematic reviews and Meta-Analyses (PRISMA) flow chart indicating the number of articles that were searched in PubMed, Scopus, and Web of Science (WoS), screened, and included in the current scoping review (ASM, artisanal and small-scale mining).

**Figure 2 ijerph-18-01555-f002:**
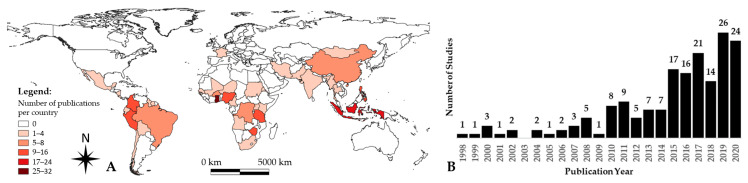
Geographical and temporal distribution of included articles (*n* = 176) per country (**A**) and year of publication (**B**). Source: authors’ compilation from extracted data; national administrative boundaries dada ware obtained at Anomaly Hotspots of Agricultural Production (ASAP) (https://mars.jrc.ec.europa.eu/asap/files/gaul0_asap_v04.zip) [46].

**Figure 3 ijerph-18-01555-f003:**
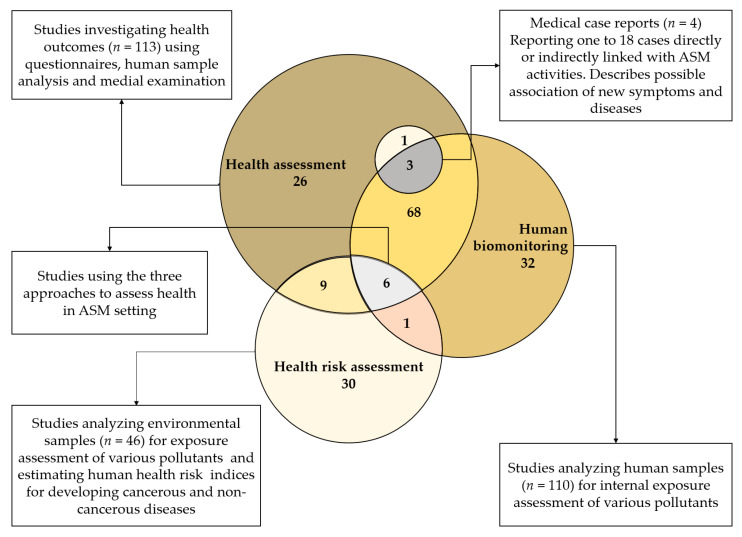
Overview of the characteristics of the 176 included health studies. Three main approaches for studying health were employed, including (i) health assessment (HA); (ii) human biomonitoring (HBM); and (iii) health risk assessment (HRA). Overlaps indicate studies for which more than one approach was used.

**Figure 4 ijerph-18-01555-f004:**
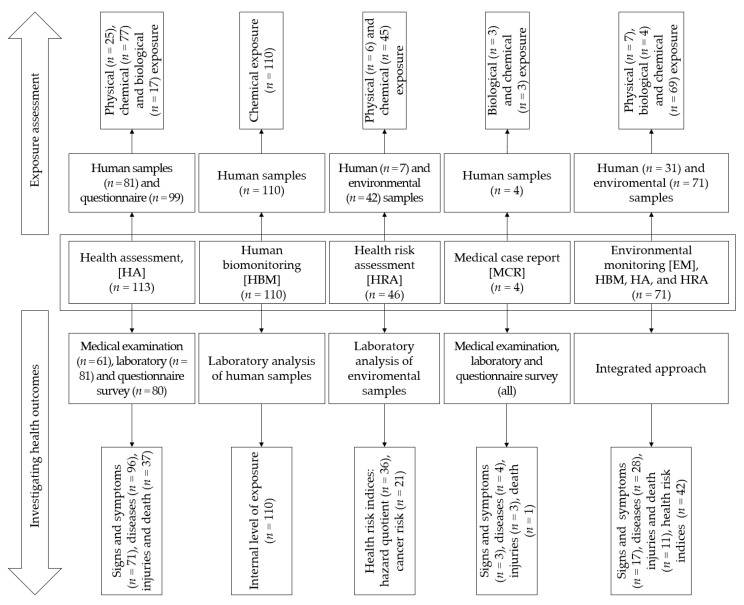
Overview of the main study type clusters, and a conceptual framework of exposure assessment and health outcome investigation strategies. Overlaps exist between different approaches; hence, frequency exceeds the total of 176 included studies.

**Figure 5 ijerph-18-01555-f005:**
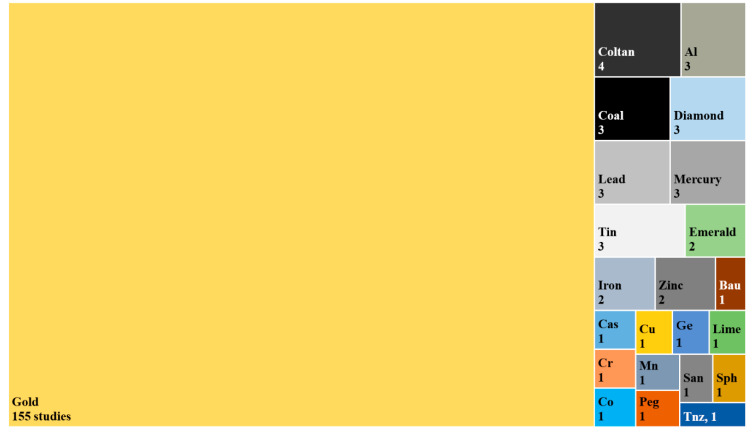
Type of ASM studied as a function of extracted commodities. The numbers indicate the total of reporting studies. Note: Al, aluminium; Bau, bauxite; Cas, cassiterite; Co, cobalt; Cr, chrome; Cu, copper; Ge, gemstones; Lime, limestones; Mn, manganese; Peg, pegmatite; San, sandstones; Sph, sphalerite; and Tnz, tanzanite.

**Figure 6 ijerph-18-01555-f006:**
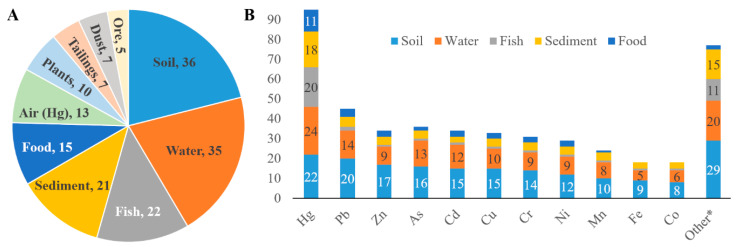
Profile of investigated environmental samples and pollutants. Numbers are frequency of reporting articles. Panel (**A**), type of examined samples and Panel (**B**), type of investigated pollutants. Note: Hg, mercury; Pb, lead; Zn, zinc; As, Arsenic; Cd, cadmium; Cu, copper; Cr, chromium; Ni, nickel; Mn, manganese; Fe, iron; Co, cobalt; and (*****) other environmental pollutants reported in few studies (e.g., 1 to 4 studies, each).

**Figure 7 ijerph-18-01555-f007:**
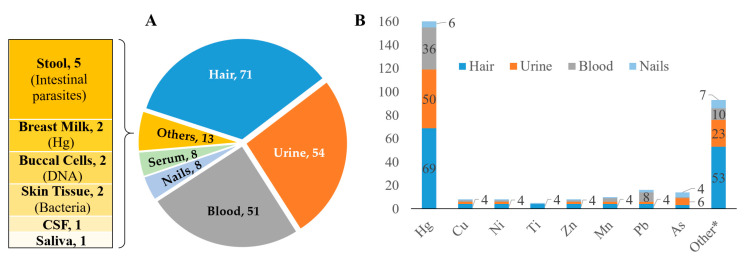
Human samples and biomarker profile. Investigated biomarkers include heavy metals and metalloids. Numbers are frequency of reporting articles. Panel (**A**), type of examined samples and Panel (**B**), type of investigated pollutants. Note: CSF, cerebrospinal fluid; DNA, deoxyribonucleic acid; Hg, mercury; Pb, lead; Zn, zinc; As, arsenic; Cu, copper; Ni, nickel; Mn, manganese; Ti, titanium; and (*****) other chemical pollutants reported in few studies (e.g., 1 to 3 studies, each).

**Figure 8 ijerph-18-01555-f008:**
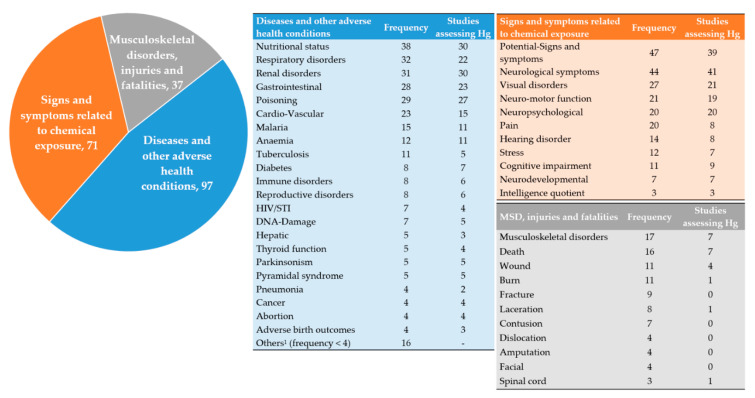
Main and sub-categories of adverse health conditions investigated at both population and individual level in ASM settings. Note: DNA, deoxyribonucleic acid; Hg, mercury, HIV, human immunodeficiency virus; MSD, musculoskeletal disorders; and STIs, sexually transmitted infections. Other adverse health outcomes are reported in less than four studies, each.

**Figure 9 ijerph-18-01555-f009:**
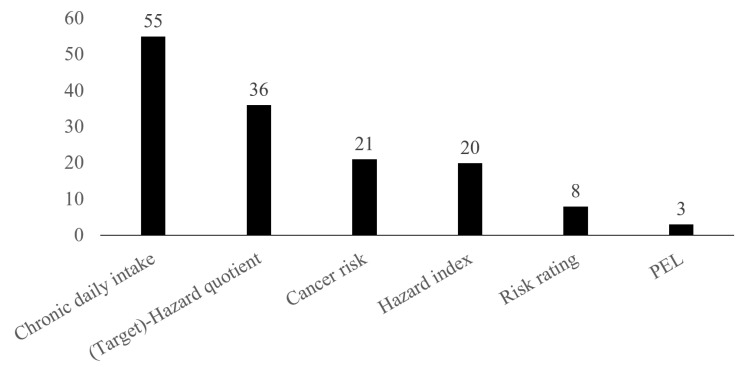
Health risk indices reported 46 out of 176 included studies. Figures represent the number of studies reporting each index. Note: PEL, permissible exposure limit.

**Table 1 ijerph-18-01555-t001:** Distribution of study populations by age group, gender, and population groups (*n* = 176).

Population Characteristics	Children (<15 Years)	Adolescents and Adults (≥15 Years)	Adults (≥18 Years)	All Ages (≥6 Months)	Age Group Not Defined	Total
**Gender**						
Males and females	8	14	50	77	5	154
Female	1	5	2	1	1	10
Male	0	0	9	1	1	11
Gender not defined	1	0	0	0	0	1
**Population groups**						
Miners and residents	1	5	21	57	1	85
Miners	0	8	32	7	5	52
Residents	9	6	8	15	1	39
Total	10	19	61	79	7	176

## Data Availability

Not applicable.

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
