# Peer review of "Health Studies in the Context of Artisanal and Small-Scale Mining: A Scoping Review"

_ijerph, 2021, doi:10.3390/ijerph18041555_

Round 1

Reviewer 1 Report

The paper is reviewing publications covering health studies in the context of artisanal and small-scale mining. Utilizing method of PRISMA ScR, the author conducted screening and filtering with some inclusion criteria then finally synthesizing, and found some research gaps. There are some points that need to answer and clarify, as follows:

  1. Too many articles are excluded during the screening process (More than 85%). How did the author, in the first step, select relevant studies resulting 438 raw hits from PubMed, 1052 from Scopus, and 1274 from WoS? Why the author did not directly focus on ASM to search articles in the three electronic databases?.
  2. Can the author elaborate the duplicate criteria that used to exclude many articles? Is that duplicate by title or by object of research?
  3. Base on the statement in line 453 – 455, this is the most important point in the way of solving ASM issues all around the world. This paper would has high impact if the author can add a sub-section discussing integrative approach not only from one reference [97].
  4. Base on the statement in line 505 – 507, mercury intoxicant is the major problem in ASM activities. One of the refence to discuss about that the title is Mercury Exposure and Health Problems in Urban Artisanal Gold Mining (UAGM) in Makassar, South Sulawesi, Indonesia https://doi.org/10.3390/geosciences7030044. The authors can considerate this reference.

Reviewer 2 Report

In this interesting paper, the literature on the health risks in artisanal and small-scale mining is a review of peer-reviewed publications. After a thorough systematic screening, 176 health studies from 38 countries were retained for final analysis. The authors conclude that when considering the global magnitude of this kind of mining practice, the number of health-related studies in artisanal and small-scale mining appear to be low and several specific contexts are not adequately represented in the literature. A majority of the articles identified is representing gold mining contexts, and thus other types of mining are largely neglected. Little attention was given to vulnerable groups, such as women of reproductive age and 27 children.

Some comments:

The paper is well written and interesting to read. Also, the tables are easy to understand and the figures are illustrative.

I suggest a clearer definition of artisanal and small-scale mining. Is artisanal mining the same as and small-scale mining?

A list of abbreviations would be very useful. The abbreviations HBM and HRA are not explained in the text, only in Figure 4.

The risk för occupational diseases in miners was already reported by Ramazzini in 1700, maybe earlier. Some historical notes would be of value in the introduction.

I am surprised by that the first paper published on this subject was published as late as 1998. What is the explanation? In the discussion, the authors conclude that health research in ASM is a neglected area, but I still think it is strange that not a single study was found before 1998. Is it explained by some factor in the inclusion criteria of studies?

Reviewer 3 Report

Manuscript, entitled " Health Studies in the Context of Artisanal and Small-Scale Mining: A Scoping Review " I recommend publishing this paper after major revision.
The manuscript is very long, I recommend shortening and accurately focused and described the main idea and findings.

The manuscript is very general and does not focus precisely on health effects in relation to artisanal and general mining. Readers expect the most detailed information on health effects
(Serious effects of metals in the study area on the population, etc.).

Suggested notes for author:
In part materials and methods - the data treatment is not described.

Figure 2 – the map is copied from ASAP and graph aslo??, is not know the origin of map and graph, the listed link not refers this map and graph

Line 204-216 – explain exactly the abbreviation (HBM, HRA, etc.) in this text part (not in Fig.3)

Fig.5 – explain the construction of this fig. (which statistical software authors used?)

In appendix, the many tables are listed and some tables are redundant.
The Table A2 is unnecessary, the list of elements and its names is generally know

The term muscosceletal means as mucosceletal?

Check references and format according the instructions for authors.

The references 73,96 have not the date of publication
